# Is Locate-then-edit Really Good at LLM Knowledge Editing? Evaluating Efficacy Beyond the Literal Edit-Target String

## Abstract

Knowledge editing refers to updating, deleting, or forgetting outdated or incorrect knowledge in large language models (LLMs). Compared to traditional methods like fine-tuning, retrieval-augmented generation or introducing extra memory modules, locate-then-edit (LTE) has recently emerged as a promising paradigm of the current literature due to its great effectiveness and efficiency: by precisely editing a small subset of parameters such that a specific fact is updated while preserving other knowledge. Despite its great success reported in previous literature, we find the apparent reliability of LTE rests on a fragile foundation and the current literature is largely driven by illusory success. Other than utilizing real semantics, the fundamental goal of steering the model's output toward a target with minimal modification could encourage exploiting hidden shortcuts, something like adversarial attack. This problem directly challenges the feasibility of the current LTE literature at its very foundation, as shortcuts are inherently at odds with robust knowledge integration. Coincidentally, this issue has long been obscured by evaluation frameworks that lack the design of negative examples. To uncover it, we systematically develop a suite of new evaluation methods. Strikingly, we find that state-of-the-art approaches collapse even under simplest negation queries. Our empirical studies uncover that LTE is likely to be based on shortcuts rather than full semantics, calling for an urgent reconsideration of the very basis of LTE before further advancements can be meaningfully pursued.

## 1 Introduction

Large language models (LLMs) inevitably encode outdated or incorrect knowledge due to their static training data. While retraining or continual learning can in principle refresh model knowledge, such approaches remain prohibitively expensive. Concise locate-then-edit (LTE) methods have therefore emerged as an attractive alternative: by precisely editing a small subset of parameters (as shown with an example in Figure 1(a)), one can supposedly update a model's knowledge with minimal cost while preserving its other knowledge. The typical paradigm operates by first precisely locating the decisive tokens in the text and the decisive layers in the model, and then replacing the hidden states of these tokens after the decisive layer, thereby enabling efficient and precise knowledge substitution (Meng et al., 2022).

The promise of efficiency and precision has inspired an impressive wave of methods (Ma et al., 2025; Li & Chu, 2025; Fang et al., 2025; Dai et al., 2025; Qiao et al., 2025; Jiang et al., 2025b; Zhai et al., 2025; Park et al., 2025), with many reporting great success rates and have quickly become one of the mainstreams of LLM knowledge editing. Building on such optimism, recent efforts have also begun to extend LTE towards complex reasoning tasks (Dong et al., 2025; Zhang et al., 2025), positioning it as a lightweight alternative to more resource-intensive paradigms such as fine-tuning or retrieval augmentation.

In this paper, however, we regret to say that the progress in this field might be too optimistic. Our analysis reveals that the apparent reliability of LTE rests on a fragile foundation. And we may even need to re-examine the basis of this field.

Deep neural networks inevitably contain shortcuts (Goodfellow et al., 2015). The fundamental paradigm of LTE is to steer the model's output towards a target with the least effort. It has been widely acknowledged that such an objective can be easily achieved by exploiting semantically meaningless adversarial shortcuts in the literature of adversarial attack[1]. In adversarial attacks, minimal changes are made to the input so that the predictor output is changed but the semantics of the input when viewed in other ways, e.g. visually, remains unchanged. Since LTE and adversarial attack share the same paradigm of steering the model's output with minimal cost, an obvious intuition is that editing may also achieve its objective through networks' hidden shortcuts. However, the purpose of editing is entirely different from that of attacking: the purpose of editing is not to destroy the model, but to enable it to acquire new semantics-based knowledge, which should not rely on attack-style shortcuts. **But is there anything can promise that the edit is done on the real semantics rather than through unknown shortcuts?** We think this is an important question that should be answered before we keep on advancing the field in a right direction. Currently, we think the answer is No, and we find empirical evidence showing that this problem consistently happens in recent LTE methods.

We first design two simple methods to expose the hidden problems into observable phenomenon. One is applying very simple negation to the test queries (see the second example in Figure 1(b) for intuition). The second one is to do the fact-checking style evaluation where the knowledge is the same but the ground truth answer is replaced from the edit target string to its proxy of "true/false" (see the third example in Figure 1(b)). For the first case, **all** (nine) state-of-the-art methods collapse entirely on **all** (four) datasets. For the second case, all methods get a significant performance drop.

We contend that this points to more than a collection of failure cases. Instead, it reveals a fundamental dilemma of LTE. The precise editing mechanism, which aggressively identifies the position that best steers the output toward the target, is overly narrow: it operates solely in the direction of when to output "Trump", without complementary guidance on other similar queries. Although this aggressive approach ensures the very precision and efficiency that make LTE appealing, it may also fall into the trap of shortcuts, akin to adversarial attacks, which conflicts with the semantic completeness required for robust knowledge integration.

Unlike issues of scaling or regularization, this tension is intrinsic to the LTE paradigm itself, challenging the foundation of current LTE literature. In summary, the contributions include:

- **Problem identification at the evaluation level**. We demonstrate that existing benchmarks systematically overlook the importance of negative cases and thereby allow shortcuts to masquerade as genuine knowledge integration.
- **Problem identification at the mechanism level**. We find that the fundamental paradigm of LTE can encourage shortcut-based adversarial behaviors rather than learning true semantics. This raises fundamental doubts about the feasibility of current LTE literature at its very basis.
- **Methodological advancement**. To move beyond case-by-case datasets, we introduce a new evaluation framework that systematically incorporates semantically complementary queries. This design exposes the fragility of current methods in a principled way and provides a foundation for more realistic assessments of editing reliability, thereby guiding future research toward more robust directions.

## 2 RELATED WORKS

**Retrieving information from an external knowledge base for inference-time knowledge editing**. One major line of research draws inspiration from Retrieval-Augmented Generation (RAG). These methods maintain an external knowledge base that stores up-to-date information and retrieve relevant content during generation. Representative works include Hartvigsen et al. (2023); Zheng et al. (2023); Zhang et al. (2024a); Jiang et al. (2024), among others. They suffer from the same limitations of RAG, such as depending on the quality of retrieval, constrained by context length, introducing additional latency and system complexity, etc. Consequently, such methods are better

---

[1] If you are not familiar with adversarial attack, please refer to Appendix A.1 for a brief introduction of adversarial examples to get a better intuition for this paper.

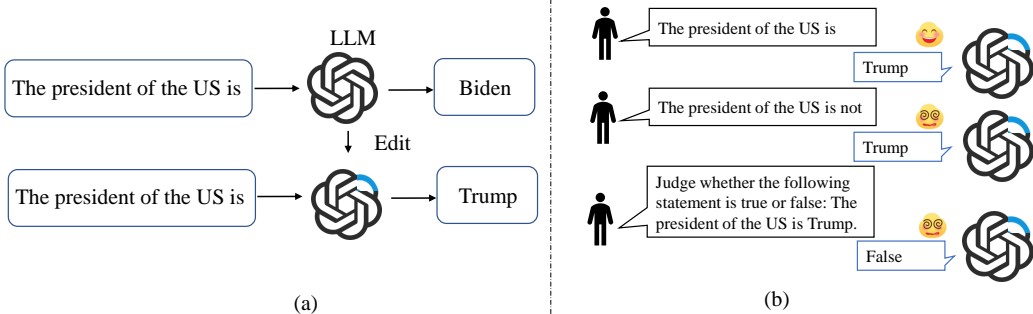

Figure 1: (a) An example of the goal of LLM editing: updating the outdated knowledge with modifying only a small set of parameters (e.g., the blue part). (b) A toy example about current paradigm of model editing is not done on the real semantics.

characterized as RAG rather than genuine model (parameter) editing, and therefore fall outside the scope of this paper.

**Training extra network memory modules or hypernetworks to store new knowledge**. Another stream of research explores augmenting models with additional memory modules that encode new knowledge. A popular strategy is to use meta-learning and hypernetworks that predict parameter updates conditioned on the edit specification. Notable examples include MEND (Mitchell et al., 2022a), KE (De Cao et al., 2021), RLEdit (Li et al., 2025) and so on, which leverage low-rank updates for efficiency. A fundamental problem of hypernetwork-based methods is that hypernetwork-based methods struggle with generalization, as different pieces of knowledge are independent, necessitating retraining/finetuning for each new fact, which is very expensive. A related line of work stores knowledge directly in auxiliary parameters or networks, as seen in T-Patcher (Huang et al., 2023), CaliNet (Dong et al., 2022), SERAC (Mitchell et al., 2022b), Grace (Hartvigsen et al., 2023), WISE (Wang et al., 2024a), KDE (Xu et al., 2025) and so on. While these methods provide flexibility, they continuously add new knowledge without discarding outdated ones. Over time, the model needs to be increasingly large, limiting scalability and practicality. Furthermore, the editing performance of this class of methods is usually weaker than that of the precise editing approaches discussed next (due to several reasons like very sensitive to hyperparameters, need a ). As a result, this family has not become the mainstream solution for knowledge editing.

**Knowledge substitution with precise model (parameter) editing**. To achieve a more elegant and parameter-efficient solution, precise model editing has recently gained momentum. These methods follow a "locate-then-edit" (LTE) paradigm: they identify a decisive token in the input and decisive layers in the model, then modify the hidden states of the decisive token to redirect the output toward the desired knowledge. Meng et al. (2022; 2023) first introduce the use of causal tracing to identify which token's (called decisive token) hidden state at which layer (called decisive layer) should be modified to most effectively steer the output toward the edit target. Subsequent works such as MEMIT (Meng et al., 2023), RECT (Gu et al., 2024), EMMET(Gupta et al., 2024), PMET(Li et al., 2024), PRUNE (Ma et al., 2025), AdaEdit (Li & Chu, 2025), AlphaEdit (Fang et al., 2025), NAMET (Dai et al., 2025), MEMIT-LTI (Zhang et al., 2025), Anyedit (Jiang et al., 2025a) build upon this paradigm by incorporating various regularization techniques or optimization strategies. These approaches have quickly become a mainstream due to their simplicity and appealing effectiveness, and recent work has extended this line of research to more complex tasks such as multi-hop reasoning (Dong et al., 2025; Zhang et al., 2025). And this line of research is the main focus of this paper.

Despite these advances, we think a critical question remains: does precise LTE inherently sacrifice semantic completeness for precision and effectiveness? By aggressively steering outputs through the quickest path, these methods may fall into the trap of potential hidden shortcuts, allowing the model to get high edit success rate with utilizing only a partial of shortest key associations between the decisive token and target output string, rather than the complete semantics.

**Complex task settings**. Some of previous research finds that current methods do not perform well on complex task settings. For example, Ma et al. (2024) find that in real-world practice where the query is rephrased or some more contexts are added in front of the query, the edited models tend

to fail. Zhang et al. (2025) show that edited models often fail on multi-hop reasoning tasks due to overfitting. Xie et al. (2025) find and explore the problem that edited models tend to reverts to its original knowledge when exposed to carefully crafted prompts. Yang et al. (2025) find that in free-form generation without teacher forcing or truncation, some (not all) editing methods experience severe performance degradation. Our position is very different from them, since we focus on the foundation of locate-then-edit, we adopt minimally complex settings in order to minimize potential confounding factors (e.g., the capacity of multi-hop relation extraction): simple datasets, simple prompts, and simple answer styles. Even on very **simple settings**, recent methods that are tested by us **all fail**.

## 3 PRELIMINARIES OF MODEL EDITING

**Notations**. Let $f_\theta$ LLM to be updated, with $\theta$ being its parameters. After editing, the updated model is written as $f_{\theta^*}$. The target knowledge set is defined as

$$S^* = \{(x_i^*, y_i^*)\}_{i=1}^n, \tag{1}$$

where $x_i^*$ is an edit input that triggers the knowledge (e.g., *The president of the US is*), and $y_i^*$ is the desired output (e.g., *Trump*), and $n$ is number of the pieces of knowledge to be rectified. To ensure that unrelated knowledge is preserved, a representative set $S = \{(x_j, y_j)\}_{j=n+1}^{n+u}$ is typically sampled from a background corpus such as Wikipedia. Note that only $x_j$ is sampled from Wiki, while $y_j$ is obtained directly from the model's current predictions.

The general goal of model editing is therefore to update specific knowledge items while leaving other model behaviors unchanged. This can be formulated as:

$$\theta^* = \arg\min_{\hat{\theta}} \left( \sum_{i=1}^n \mathcal{L}_1(f_{\hat{\theta}}(x_i^*), y_i^*) + \lambda \sum_{j=n+1}^{n+u} \mathcal{L}_2(f_{\hat{\theta}}(x_j), y_j) \right), \tag{2}$$

where $\mathcal{L}_1$ and $\mathcal{L}_2$ are some loss functions for the edit and preservation objectives, respectively, and $\lambda$ balances the two.

**Model editing**. Editing all parameters in an LLM is computationally prohibitive. To address this, Meng et al. (2022) propose the causal tracing method to identify decisive tokens and decisive layers. Specifically:

- **Decisive token**: the token whose hidden representation most strongly determines the factual output (often the last token of the subject in the query)[2].

- **Decisive layer**: the layer at which modifying the hidden state of this token most effectively steers the model toward the edit target.

By intervening only on the hidden state of the decisive token at the decisive layer, model editing can selectively overwrite factual associations with high efficiency and precision. This paradigm has rapidly become the standard due to its simplicity and strong empirical performance.

**A Canonical Objective**. Although methods vary in implementation, most can be expressed as variants of the following optimization problem:

$$\Delta^* = \arg\min_{\Delta} \left( \sum_{i=1}^n \|(W+\Delta))k_i - m_i\|_2 + \lambda \sum_{i=n+1}^{n+u} \|(W+\Delta)k_i - m_i\|_2 \right),$$
$$W^* = W + \Delta^* \tag{3}$$

where $W$ is a subset of model parameters, $k_i$ is the hidden state of the decisive token (the last token of the subject) before the decisive layer, and $m_i$ is the idealized hidden state aligned with the edit target. In this paper, we focus on how the target knowledge is updated and do not discuss how unrelated knowledge is maintained, so we will omit the second part of Eq. 3 for simplicity of presentation in the following sections: when we say Eq. 3, we are actually referring to the first term $\arg\min_{\Delta} \sum_{i=1}^n \|(W+\Delta))k_i - m_i\|_2$

---

[2]This phenomenon has been investigated across multiple studies (Meng et al., 2022; Xie et al., 2025) and is now widely regarded as a consensus within the field.

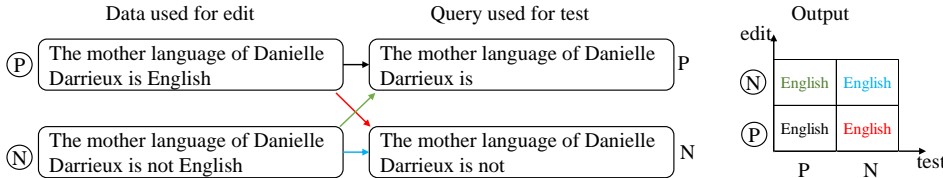

Figure 3: A qualitative illustrative example of the experimental failure case under negation (old model knowledge is French, see Figure 2). Either the model is edited with "XX is YY" or "XX is not YY", and either the test query is "XX is" or "XX is not", results consistently tend to be "YY".

Consider the example in Figure 2, $x$ is "The mother language of Danielle Darrieux is", then $k_i$ should be the hidden state of "Darrieux" at the layer before $W$. As for $m_i$, it is the idealized hidden state of "Darrieux" at the layer after $W$. In other words, if we change the output of $W$ from $Wk$ to $m_i$, then the LLM can change the output from the old knowledge (e.g., "French") to the edit target (e.g., "English"). In practice, gradient descent is used to find $m_i$, after which parameter updates are computed to ensure that $(W + \Delta)k_i$ maps closely to $m_i$.

# 4 OUR HYPOTHESES ON THE RISKS OF EXISTING WORKS

Despite the fact that model editing is a popular field and is growing rapidly, we caution that it may be advancing on a fragile foundation, driven by illusory successes.

Deep neural networks inevitably contain shortcuts. Borrowing from the philosophy of targeted adversarial attacks, it is often easy to find semantically meaningless shortcuts that can drive the model to produce target outputs. The basic idea of model editing is to find the decisive position that steers the output toward the target with least efforts. **But is there anything that can promise that the edit is done on the real semantics rather than through unknown shortcuts?** We think this is an important question that should be answered before we keep on advancing the field. Currently, we think the answer is No.

**A data example for the sake of presentation**. Here we provide a data example from the Counterfact dataset to make the following presentation easier. We will have an input query, an old knowledge output, and an edit target. The input query is what we send to the models, and the old knowledge output is what the pre-edited model will output, and the edit

> **Input query**: The mother language of Danielle Darrieux is
> **Old knowledge**: French
> **Edit target**: English

Figure 2: A data example from Counterfact.

target is what we want the edited model to output. An example is shown in Figure 2, it represents a scenario where we want to rectify the incorrect old knowledge "French" to new target "English".

## 4.1 EMBARRASSING FAILURE BY SIMPLE NEGATION

We select a set of powerful methods published recently which can perform very well on updating "French" with "English". However, the edited models are also very confident to output "English" when facing the negation query "The mother language of Danielle Darrieux is not". Please refer to §5.2 for more details and results.

The design to expose the above failure case stems from our hypothesis that current model editing encourages shortcut learning rather than genuine semantic understanding. The objective of Eq. 3 primarily focuses on the decisive token and the edit target (e.g., "Darrieux" and "English"), while other supportive tokens in the context receive little attention. Our hypothesis is whether Eq. 3 merely encourages an association between the decisive token and the edit target (e.g., Darrieux" and English"), while supportive tokens in between contribute little to the edit? Although information can propagate along the input sequence such that modifications to the hidden state of a decisive token may also influence supportive tokens, this influence is indirect and incidental. Such incidental effects may not be strong enough to alleviate token-level associations.

To further explore how the supportive tokens work, we design a verification from the opposite of the above failure case. We change all the input query in the edit set into its negation and use this set to edit the model (Figure 3, see §5.2 for more details). There are in total four experiments (the following is only an illustrative example for intuitive understanding; in practice, our analysis is based on statistical experiments over the dataset): ① PP (positive edit & positive test): We use "The mother language of Danielle Darrieux is English" as the data for editing, and use "The mother language of Danielle Darrieux is" as the test query. ② PN (positive edit & negative test): We use "The mother language of Danielle Darrieux is English" as the data for editing, and use "The mother language of Danielle Darrieux is not" as the test query. ③ NN (negative edit & negative test): We use "The mother language of Danielle Darrieux is not English" as the data for editing, and use "The mother language of Danielle Darrieux is not" as the test query. ④ NP (negative edit & positive test): We use "The mother language of Danielle Darrieux is not English" as the data for editing, and use "The mother language of Danielle Darrieux is" as the test query. We find that in all these four cases, edited models consistently output "English".

**Remark**. The results indicate that whether the model is edited with "XX is YY" or "XX is not YY", and whether the test query is "XX is" or "XX is not", the outputs show little difference across the four cases. The supportive tokens ("is"/"is not") seem to have little influence, neither at editing time nor test time.

It seems that, the mechanism of precisely focusing on the decisive hidden state that most effectively steers the output toward the edit target can encourage shortcuts by bypassing the utilization of the real semantics. The reason is that locating follows an "attack"-style strategy: it operates solely in the offensive direction of when to output "English", without any complementary guidance on the defensive side of when not to do so. As a result, it does not have the incentive to leverage the full semantics or to capture the subtle distinctions provided by supportive tokens. That is why we believe precise edit and semantic completeness are conflicting goals, challenging the feasibility of current model editing.

Furthermore, it is clear that the current literature faces a critical issue concerning the evaluation framework: existing edit success rate designs lack the incorporation of negative cases, a shortcoming that masks the problem of methods achieving illusory high scores through shortcuts.

## 4.2 Failure Case 2: Fact-checking Style Evaluation

Aside from the absence of negative cases, the current evaluation framework has another flaw that allows shortcut-based success to remain hidden. In current literature regarding the edit success evaluation, the ground truth is always the tokens of the edit target. The question is: what if the input query conveys the same knowledge, but the ground truth answer is not identical to the edit target tokens—i.e., the semantics are the same but the surface form differs? For example, consider evaluating the edited model's ability to perform fact-checking on whether "The mother language of Danielle Darrieux is English" is true or false (see §5.3 for details).

We find that all methods get a significant performance drop on this simple fact-checking style evaluation even if they get very high success rate when the ground truth is just the edit target. Since the two evaluation tasks are roughly comparable in difficulty, such a large performance gap would not be expected. The large discrepancy in success rates between these two settings further supports the possibility that model editing is encouraging shortcuts rather than the utilization of real semantics.

## 5 Statistical Experiments

## 5.1 Experimental Setup

**Base LLMs and model editing methods**. We employ two popular open-source models widely used in previous model editing literature: Qwen2.5-7B-Instruct and Llama-3-8B-Instruct. We employ nine recent methods that have achieve great performance under previous evaluation methods, including MEMIT (Meng et al., 2023), RECT (Gu et al., 2024), EMMET(Gupta et al., 2024), PMET(Li et al., 2024), PRUNE (Ma et al., 2025), AdaEdit (Li & Chu, 2025), AlphaEdit (Fang et al., 2025), NAMET (Dai et al., 2025), MEMIT-LTI (Zhang et al., 2025). Among them, LTI is a method aims to address overfitting in model editing. We use this baseline to show that the problem found by us

Table 1: An example of the experimental designs of §5.2

| Date used for edit | Test query |
| --- | --- |
| The mother language of Danielle Darrieux is English | The mother language of Danielle Darrieux is |
| The mother language of Danielle Darrieux is English | The mother language of Danielle Darrieux is not |
| The mother language of Danielle Darrieux is not English | The mother language of Danielle Darrieux is not |
| The mother language of Danielle Darrieux is not English | The mother language of Danielle Darrieux is |

is not a simple overfitting problem. We also include a recent method that does not belong to locate-then-edit: RLEdit (Li et al., 2025) (a hypernetwork-based method). RLEdit is implemented on only MCF and ZsRE datasets, because it needs a training set but it only provides the training set for MCF and ZsRE. Also, RLEdit results do not include $NN$ and $NP$ settings because building a negative training set for it is expensive.

**Datasets**. We employ four widely used datasets, including Multi-Counterfact (MCF) (Meng et al., 2022), ZsRE (Levy et al., 2017), MQuAKE (Zhong et al., 2023), and Wiki-Counterfact (WCF) (Zhang et al., 2024b). We employ the Efficacy score (edit success rate) (Meng et al., 2023; Fang et al., 2025) widely use in previous literature as the basic metric. More special metrics proposed by us will be introduced in the experimental designs. We use an H100-80G GPU to edit the models.

**Task setting and baseline implementation**. Most of the settings are copied from the recent well-known paper AlphaEdit. We follow AlphaEdit to consider the scenario that combines both sequential editing and batch editing: For each dataset, we edit 2000 samples, with 100 samples per edit batch. The decisive layers for base models are from a popular public repository: `https://github.com/zjunlp/EasyEdit` (Wang et al., 2024b). We note that for the baseline methods, we implement them with our improved version. Specifically, since AlphaEdit finds that in sequential editing, performance can be greatly improved by including previously edited knowledge from earlier batches into the model's retained knowledge set when editing subsequent batches, we also apply this technique to other baseline methods to enhance their overall performance.

## 5.2 NEGATION OF KNOWLEDGE QUERIES

**Design**. In this part, we want to explore how much the supportive tokens between the decisive token and edit target work. For each knowledge edit, we construct a simple negated form and systematically combine the positive and negative variants of the edit data with the corresponding test queries. This yields four experimental settings, as illustrated with a concrete example in Table 1 and Figure 3. We denote these settings as ① **p**ositive edit & **p**ositive test (PP), ② positive edit & negative test (PN), ③ negative edit & negative test (NN), ④ negative edit & positive test (NP). For all four cases, we uniformly treat the edit target (e.g., English) as the ground truth for computing the **Efficacy** score (i.e., edit success rate, please refer to Appendix A.2 for more details). In prior work, the edit success rate is typically defined as a successful edit when the probability of generating the edit target exceeds that of generating the original knowledge (Meng et al., 2023; Fang et al., 2025). In contrast, we adopt the criterion by requiring the model output matches the edit target itself to count as a success (see Appendix A.2). Of these four settings, PP corresponds to the evaluation protocol commonly used in the literature, whereas the remaining settings are introduced in this work for the first time. We note that under the PN and NP settings, there is no gold ground-truth answer. However, since the correct answer should not be the edit target, we can still treat the edit target as the ground truth to compute a special "Efficacy" score, which should be appropriately interpreted as a "**Hallucination**" score. Discrepancy represents the difference obtained by subtracting the Hallucination from Efficacy. We define the outputs obtained by using semantically opposite queries during

Table 2: Results on Llama3-8B-Instruct. The metric for efficacy is exact match. "No edit": the original LLM without editing, and the ground truth for it is not the edit target but the old model knowledge. ZsRE is not a counterfactual dataset so it does not have the No edit results. See Appendix A.7 for details.

| Metrics | | Efficacy↑ / Hallucination↓ | | | | Discrepancy (Rectified Efficacy) | | | | |
|---|---|---|---|---|---|---|---|---|---|---|
| Methods | | PP ↑ | PN ↓ | NN ↑ | NP ↓ | PP–PN↑ | PP–NP↑ | NN–PN↑ | NN–NP↑ | **Avg↑** |
| MCF | No edit* | 100 | 15.4 | - | - | 84.6 | - | - | - | 84.6 |
| | RLEdit | 17.8 | 15.2 | - | - | 2.6 | - | - | - | 2.6 |
| | Adaedit | 97.4 | 76.3 | 97.3 | 86.4 | 21.1 | 11.0 | 21.0 | 10.9 | 16.0 |
| | AlphaEdit | 95.6 | 73.1 | 96.0 | 83.0 | 22.5 | 12.6 | 22.9 | 13.0 | 17.8 |
| | EMMET | 97.0 | 76.7 | 96.9 | 84.6 | 20.3 | 12.4 | 20.2 | 12.3 | 16.3 |
| | MEMIT | 98.2 | 69.4 | 97.8 | 81.5 | 28.8 | 16.7 | 28.4 | 16.3 | 22.6 |
| | NAMET | 98.4 | 70.2 | 97.8 | 81.4 | 28.2 | 17.0 | 27.6 | 16.4 | 22.3 |
| | PMET | 97.0 | 76.0 | 97.0 | 86.4 | 21.0 | 10.6 | 21.0 | 10.6 | 15.8 |
| | PRUNE | 97.8 | 69.4 | 97.8 | 81.6 | 28.4 | 16.2 | 28.4 | 16.2 | 22.3 |
| | RECT | 98.4 | 69.2 | 97.8 | 81.0 | 29.2 | 17.4 | 28.6 | 16.8 | 23.0 |
| | MEMIT-LTI | 97.7 | 66.9 | 97.0 | 75.6 | 30.8 | 22.1 | 30.1 | 21.4 | 26.1 |
| ZsRE | RLEdit | 28.2 | 25.2 | - | - | 3.0 | - | - | - | 3.0 |
| | Adaedit | 95.6 | 88.6 | 95.0 | 89.7 | 7.0 | 5.9 | 6.4 | 5.3 | 6.2 |
| | AlphaEdit | 94.2 | 87.5 | 93.6 | 86.8 | 6.7 | 7.4 | 6.1 | 6.8 | 6.8 |
| | EMMET | 95.4 | 87.3 | 95.2 | 87.8 | 8.1 | 7.6 | 7.9 | 7.4 | 7.7 |
| | MEMIT | 95.5 | 87.0 | 94.8 | 83.5 | 8.5 | 12.0 | 7.8 | 11.3 | 9.9 |
| | NAMET | 95.6 | 86.4 | 94.7 | 82.8 | 9.2 | 12.8 | 8.3 | 11.9 | 10.6 |
| | PMET | 96.4 | 87.6 | 95.7 | 91.9 | 8.8 | 4.5 | 8.1 | 3.8 | 9.4 |
| | PRUNE | 95.2 | 85.8 | 94.8 | 84.1 | 9.4 | 11.1 | 9.0 | 10.7 | 10.1 |
| | RECT | 95.1 | 86.0 | 94.2 | 81.9 | 9.1 | 13.2 | 8.2 | 12.3 | 10.7 |
| | MEMIT-LTI | 93.6 | 83.0 | 93.6 | 87.0 | 10.6 | 6.6 | 10.6 | 6.6 | 8.6 |
| WCF | No edit* | 100 | 0.8 | - | - | 99.2 | - | - | - | 99.2 |
| | Adaedit | 73.9 | 8.0 | 69.2 | 53.2 | 65.9 | 20.7 | 61.2 | 16.0 | 41.0 |
| | AlphaEdit | 65.4 | 4.2 | 64.2 | 36.2 | 61.2 | 29.2 | 60.0 | 28.0 | 44.6 |
| | EMMET | 65.1 | 8.2 | 59.4 | 42.9 | 56.9 | 22.2 | 51.2 | 16.5 | 36.7 |
| | MEMIT | 77.5 | 6.2 | 73.7 | 43.6 | 71.3 | 33.9 | 67.5 | 30.1 | 50.7 |
| | NAMET | 78.3 | 6.2 | 73.5 | 46.8 | 72.1 | 31.5 | 67.3 | 26.7 | 49.4 |
| | PMET | 73.8 | 7.2 | 70.4 | 55.8 | 66.6 | 18.0 | 63.2 | 14.6 | 40.6 |
| | PRUNE | 78.0 | 5.4 | 73.8 | 45.6 | 72.6 | 32.4 | 68.4 | 28.2 | 50.4 |
| | RECT | 76.6 | 5.0 | 73.7 | 43.0 | 71.6 | 33.6 | 68.7 | 30.7 | 51.2 |
| | MEMIT-LTI | 69.8 | 5.4 | 66.9 | 34.2 | 64.4 | 35.6 | 61.5 | 32.7 | 48.6 |
| MQuAKE | No edit* | 100 | 11.5 | - | - | 84.6 | - | - | - | 88.5 |
| | Adaedit | 91.4 | 63.3 | 91.8 | 81.8 | 28.1 | 9.6 | 28.5 | 10.0 | 19.1 |
| | AlphaEdit | 91.4 | 53.4 | 89.8 | 73.2 | 38.0 | 18.2 | 36.4 | 16.6 | 27.3 |
| | EMMET | 77.9 | 52.4 | 77.2 | 67.3 | 25.5 | 10.6 | 24.8 | 9.9 | 16.4 |
| | MEMIT | 96.7 | 60.0 | 95.6 | 79.0 | 36.7 | 17.7 | 35.6 | 16.6 | 26.7 |
| | NAMET | 96.0 | 59.2 | 95.6 | 79.0 | 36.8 | 17.0 | 36.4 | 16.6 | 26.6 |
| | PMET | 91.3 | 64.9 | 92.0 | 81.5 | 26.4 | 9.8 | 27.1 | 10.5 | 18.5 |
| | PRUNE | 96.2 | 60.2 | 95.9 | 78.4 | 36.0 | 17.8 | 35.7 | 17.5 | 26.8 |
| | RECT | 96.4 | 60.3 | 95.4 | 78.8 | 36.1 | 17.6 | 35.1 | 16.6 | 26.4 |
| | MEMIT-LTI | 94.6 | 56.9 | 92.8 | 76.0 | 37.7 | 18.6 | 35.9 | 16.8 | 27.3 |

editing and testing as hallucination. We then compute **rectified efficacy** as the difference between efficacy and hallucination.

If a method is genuinely capable of injecting new knowledge into the model, we would expect the completely opposite combinations of edit data and test queries (PN and NP) to yield very low scores. For example, in the NP and PN settings, the edited model should not output English, and therefore their scores are expected to be low. Surprisingly, the results were striking: the performance across PP, PN, NN, and NP settings differed only marginally.

**Results**. The results with Llama3-8B-Instruct are shown in Table 2. We see that, except RLEdit (which does not belong to LTE methods), all LTE methods achieve very high scores on the PP and NN settings (where the edited knowledge and test queries are consistent), demonstrating that locate-then-edit indeed has strong capability in altering LLM behavior. However, we challenge whether this alteration truly corresponds to injecting new knowledge. By examining the PN and NP settings, we find that their scores are also high, implying that even when the edit data and test queries describe completely opposite semantics, the model still tends to output the edit target at test time.

Table 3: Results with Qwen2.5-7B-Instruct. The efficacy is calculated with exact token match.

| Methods | Metrics | Efficacy↑ / Hallucination↓ | | | | Discrepancy (Rectified Efficacy) | | | | |
|---|---|---|---|---|---|---|---|---|---|---|
| | | PP ↑ | PN ↓ | NN ↑ | NP ↓ | PP–PN↑ | PP–NP↑ | NN–PN↑ | NN–NP↑ | **Avg↑** |
| | No edit* | 100 | 16.3 | - | - | 83.7 | - | - | - | 83.7 |
| MCF | Adaedit | 84.0 | 64.8 | 80.8 | 66.8 | 19.2 | 17.2 | 16.0 | 14.0 | 16.6 |
| | AlphaEdit | 86.0 | 63.4 | 93.0 | 79.6 | 22.6 | 6.4 | 29.6 | 13.4 | 18.0 |
| | EMMET | 87.5 | 65.0 | 65.2 | 63.1 | 22.5 | 24.4 | 0.2 | 2.1 | 12.3 |
| | MEMIT | 93.4 | 72.2 | 84.9 | 74.4 | 21.2 | 19.0 | 12.7 | 10.5 | 15.9 |
| | NAMET | 93.2 | 73.0 | 88.2 | 78.5 | 20.2 | 14.7 | 15.2 | 9.7 | 15.0 |
| | PMET | 84.4 | 66.6 | 77.0 | 64.8 | 17.8 | 19.6 | 10.4 | 12.2 | 15.0 |
| | PRUNE | 93.3 | 72.6 | 93.4 | 81.9 | 20.7 | 11.4 | 20.8 | 11.5 | 16.1 |
| | RECT | 93.5 | 74.5 | 85.8 | 76.7 | 19.0 | 16.8 | 11.3 | 9.1 | 14.1 |
| | MEMIT-LTI | 65.0 | 52.2 | 66.2 | 65.0 | 12.8 | 0.0 | 14.0 | 1.2 | 7.0 |
| ZsRE | Adaedit | 87.5 | 83.4 | 88.6 | 89.7 | 4.1 | -2.2 | 5.2 | -1.1 | 1.5 |
| | AlphaEdit | 82.4 | 71.8 | 68.7 | 75.8 | 10.6 | 6.6 | -3.1 | -7.1 | -1.6 |
| | EMMET | 84.4 | 75.1 | 77.6 | 78.7 | 9.3 | 5.7 | 2.5 | -1.1 | 4.1 |
| | MEMIT | 96.5 | 88.5 | 92.8 | 90.8 | 8.0 | 5.7 | 4.3 | 2.0 | 5.0 |
| | NAMET | 94.6 | 87.7 | 94.1 | 93.2 | 6.9 | 1.4 | 6.4 | 0.9 | 2.9 |
| | PMET | 88.4 | 86.7 | 86.8 | 87.2 | 1.7 | 1.2 | 0.1 | -0.4 | 0.7 |
| | PRUNE | 95.8 | 89.4 | 92.0 | 88.8 | 6.4 | 7.0 | 2.6 | 3.2 | 5.7 |
| | RECT | 95.6 | 87.9 | 94.6 | 92.1 | 7.7 | 3.5 | 6.7 | 2.5 | 5.1 |
| | MEMIT-LTI | 62.8 | 53.2 | 35.8 | 36.0 | 9.6 | 26.8 | -17.4 | -0.2 | 4.7 |
| | No edit* | 100 | 1.1 | - | - | 98.9 | - | - | - | 98.9 |
| WCF | Adaedit | 54.8 | 9.4 | 47.9 | 42.4 | 45.4 | 12.4 | 38.5 | 5.5 | 25.5 |
| | AlphaEdit | 42.0 | 9.2 | 34.0 | 27.6 | 32.8 | 14.4 | 24.8 | 6.4 | 20.5 |
| | EMMET | 10.8 | 7.0 | 14.1 | 4.0 | 3.8 | 6.8 | 7.1 | 10.1 | 7.0 |
| | MEMIT | 58.0 | 14.8 | 43.6 | 39.4 | 43.2 | 18.6 | 28.8 | 4.2 | 24.2 |
| | NAMET | 58.0 | 12.0 | 38.0 | 33.6 | 46.0 | 24.4 | 26.0 | 4.4 | 25.2 |
| | PMET | 51.5 | 9.8 | 43.4 | 40.2 | 41.7 | 11.3 | 33.6 | 3.2 | 22.1 |
| | PRUNE | 56.6 | 12.8 | 40.3 | 37.2 | 43.8 | 19.4 | 27.5 | 3.1 | 23.5 |
| | RECT | 61.6 | 11.6 | 42.8 | 37.8 | 50.0 | 23.8 | 31.2 | 5.0 | 27.5 |
| | MEMIT-LTI | 6.4 | 02.2 | 11.3 | 4.2 | 4.2 | 2.2 | 9.1 | 7.1 | 5.7 |
| | No edit* | 100 | 11.2 | - | - | 88.8 | - | - | - | 88.8 |
| MQuAKE | Adaedit | 65.7 | 47.0 | 68.4 | 64.2 | 18.7 | 1.5 | 21.4 | 4.2 | 11.5 |
| | EMMET | 38.2 | 25.7 | 33.4 | 36.2 | 12.5 | 2.0 | 7.7 | -2.8 | 3.4 |
| | MEMIT | 69.9 | 45.6 | 43.0 | 43.9 | 24.3 | 26.0 | -2.6 | -0.9 | 10.4 |
| | NAMET | 72.4 | 51.4 | 48.4 | 48.2 | 21.0 | 24.2 | -3.0 | 0.2 | 20.0 |
| | PMET | 68.2 | 48.2 | 58.5 | 55.9 | 20.0 | 12.3 | 10.3 | 2.6 | 11.5 |
| | PRUNE | 71.0 | 50.3 | 57.8 | 56.6 | 20.7 | 14.4 | 7.5 | 1.2 | 12.5 |
| | RECT | 70.3 | 54.4 | 43.6 | 41.9 | 15.9 | 28.4 | -10.8 | 1.7 | 8.8 |
| | MEMIT-LTI | 57.0 | 38.4 | 24.3 | 28.6 | 18.6 | 28.4 | -14.1 | -4.3 | 7.2 |

The discrepancies between PP and PN (PP–PN in Table 2) as well as between PP and NP are very small. The low discrepancy between PP and PN indicates that the supportive tokens "is" / "is not" contribute very little at test time. Similarly, the low discrepancy between PP and NP suggests that these supportive tokens also play little role at edit time. The discrepancies between NN and NP, as well as between NN and PN, follow the same pattern.

As shown in our results, the rectified efficacy of all methods remains very low. This suggests that most of the observed gains are in fact illusory success, achieved primarily through shortcuts rather than genuine knowledge editing. We also see that the hallucination scores on the WCF dataset are usually higher than other datasets, but at the same time, the efficacy scores are also lower than other datasets. This further confirms the inherent tension between aggressively steering the output and leveraging genuine semantic knowledge.

We also provide the results conducted with Qwen2.5-7B-Instruct in Table 3. The phenomenon is similar, so we do not separately discuss it again.

**Conclusions**. Taken together, these results suggest that the mechanism of current model editing is likely overly aggressive, causing the utilization of shortcuts rather than real semantics. And beyond the techniques themselves, it is clear that the evaluation of edit success rate urgently requires the incorporation of negative case designs to avoid driven by deceptive success.

Table 4: Results of fact-checking style evaluation with Qwen2.5-7B-Instruct. "No edit": the original LLM without editing. See Appendix A.7 for more details of this baseline.

| | Datasets | MCF | | ZsRE | | WCF | | MQuAKE | |
|---|---|---|---|---|---|---|---|---|---|
| | Methods | Eff ↑ | Acc ↑ | Eff ↑ | Acc ↑ | Eff ↑ | Acc ↑ | Eff ↑ | Acc ↑ |
| Qwen2.5-7B-Instruct | No edit* | 100 | 91.0 | - | - | 100 | 91.5 | 100 | 93.0 |
| | Adaedit | 84.0 | 29.9 | 87.5 | 55.4 | 54.8 | 14.2 | 65.7 | 16.3 |
| | Alphaedit | 86.0 | 31.2 | 82.4 | 47.3 | 42.0 | 16.8 | 14.2 | 16.8 |
| | EMMET | 87.5 | 27.0 | 84.4 | 45.6 | 10.8 | 14.7 | 38.2 | 74.1 |
| | MEMIT | 93.4 | 37.3 | 96.5 | 48.8 | 58.0 | 13.6 | 69.9 | 14.9 |
| | NAMET | 93.2 | 36.3 | 94.6 | 49.9 | 58.0 | 14.5 | 72.4 | 16.0 |
| | PMET | 84.4 | 28.7 | 88.4 | 52.8 | 51.5 | 12.9 | 68.2 | 11.9 |
| | PRUNE | 93.3 | 32.7 | 95.8 | 49.6 | 56.6 | 14.8 | 71.0 | 15.4 |
| | RECT | 93.5 | 35.2 | 95.6 | 48.7 | 61.6 | 14.3 | 70.3 | 14.6 |
| | MEMIT-LTI | 65.0 | 28.2 | 62.8 | 48.7 | 6.4 | 18.8 | 57.0 | 15.5 |

## 5.3 FACT-CHECKING STYLE EVALUATION

We further design an alternative approach to expose that the mechanism of model editing is overly aggressive from a different perspective. We want to see what will happen when the edited model is queried with the same knowledge but the tokens of edit target no longer appears in the gold answer. To avoid increasing the difficulty of understanding the modified inputs, we adopt a simple strategy: we concatenate the original input query with the edit target to form a statement, and then ask the model to perform a fact-checking task. For example, given the input query "The mother language of Danielle Darrieux is" and the edit target "English", the corresponding test input becomes "Judge whether the following statement is true or false: The mother language of Danielle Darrieux is English". We report accuracy for this experiment. More details are in Appendix A.3.

The results with Qwen2.5 are shown in Table 4 (Llama results are in Appendix A.4). We see that, there is a significant discrepancy between the Efficacy score and the fact-checking accuracy. Further support the idea that the model editing is too aggressive to consider the real semantics during injecting new knowledge.

## 6 CONCLUSION

In this work, we first identify a critical gap in the evaluation of model editing: the absence of negative examples. We then show that this absence masks the reliance on shortcuts, and propose tailored evaluation methods (e.g., negation, fact-checking) to expose these issues. Our analysis suggests that the LTE's pursuit of higher edit success rates has become overly aggressive, to the point where the prevailing paradigm of LTE is similar to adversarial attacks. Strikingly, we find that even state-of-the-art LTE methods often achieve their reported success by exploiting shortcuts rather than by semantically integrating new knowledge. These findings highlight an urgent need to reconsider the feasibility of the LTE paradigm.

We agree that comprehensive evaluation is essential for steering the field forward. Although our proposed tests already reveal the lack of semantic grounding in current LTE methods, we take it as a starting point. Passing our tests does not necessarily promise that an edit is truly grounded in real semantics. Future work will aim to design more rigorous and holistic evaluation frameworks to better assess whether edits truly rely on real semantics.

As for the solution about how to avoid this problem, we think that more firm and balanced approaches, such as RAG-style inference-time knowledge editing, can help. Because they do not aggressively steer the output; instead, they simply provide helpful context. And fine-tuning the model with data augmentation might also avoid this problem because complementary knowledge can be absorbed from multiple data samples. However, they also face their own limitations, such as dependence on retrieval quality, additional latency, system complexity (for inference-time editing), and lack of high-quality data (for fine-tuning). In the future, besides the LTE methods, we may consider giving more attention to improving these methods as well.

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

Figure 4: The example of steering the output to "gibbon" with shortcuts.

Table 5: Results of fact-checking style evaluation with Llama3-8B-Instruct. The efficacy is calculated with exact token match.

| | Datasets | MCF | | ZsRE | | WCF | | MQuAKE | |
|---|---|---|---|---|---|---|---|---|---|
| Methods | | Eff ↑ | Acc ↑ | Eff ↑ | Acc ↑ | Eff ↑ | Acc ↑ | Eff ↑ | Acc ↑ |
| | No edit | 100 | 91.1 | - | - | 100 | 91.3 | 100 | 93.1 |
| Llama3-8B-Instruct | Adaedit | 97.4 | 68.1 | 95.6 | 62.3 | 73.9 | 62.5 | 91.4 | 31.9 |
| | AlphaEdit | 95.6 | 76.5 | 94.2 | 70.9 | 65.4 | 65.8 | 91.4 | 33.5 |
| | EMMET | 97.0 | 46.1 | 95.4 | 64.6 | 65.1 | 64.1 | 77.9 | 16.3 |
| | MEMIT | 98.2 | 64.6 | 95.4 | 64.4 | 77.5 | 60.9 | 96.7 | 32.8 |
| | NAMET | 98.4 | 62.1 | 95.6 | 64.9 | 78.3 | 58.0 | 96.0 | 32.4 |
| | PMET | 97.0 | 69.1 | 96.4 | 68.8 | 73.8 | 64.6 | 91.3 | 34.2 |
| | PRUNE | 97.8 | 63.8 | 95.2 | 64.7 | 78.0 | 62.4 | 96.2 | 31.6 |
| | RECT | 98.4 | 64.2 | 95.1 | 63.2 | 76.6 | 61.2 | 96.4 | 32.1 |
| | MEMIT-LTI | 97.7 | 54.4 | 93.6 | 60.0 | 69.8 | 45.8 | 94.6 | 20.8 |

## A   MORE RESULTS AND DETAILS

### A.1   PRELIMINARIES OF ADVERSARIAL ATTACK

Adversarial attacks have long been a central topic in deep learning research (Goodfellow et al., 2015). The key observation is that deep neural networks inevitably contain shortcuts. For example, if we want to change an image classifier's prediction from "panda" to "gibbon", there are two possible approaches. The first is straightforward: we modify the panda image into an actual gibbon image, thereby altering the true semantics. The second approach exploits shortcuts within the model: by identifying these vulnerabilities, we can add small, semantically meaningless perturbations to the panda image and still force the classifier to output "gibbon".

Figure 4 illustrates such an example. The core idea behind adversarial attacks is strikingly similar to that of model editing: both aim to steer the model's output toward a target using minimal modifications. The primary difference lies in where the modification occurs—on the input for adversarial attacks, and on the model parameters for editing. The success of adversarial attack shows that there are shortcuts allow the modification does not need to rely on the real semantics.

### A.2   EVALUATION METRICS

**Metrics about the edit success rate**. In this paper, we focus exclusively on whether the target knowledge has been correctly updated, without considering the preservation of unrelated knowledge.

We measure the edit success rate using an exact token match criterion: an edit is deemed successful only if the model's generated answer exactly matches the token sequence of the edit target.

Let $n_s$ denote the number of successful edits, and $n_t$ the number of total edits, the edit success rate (efficacy) is then computed as

$$\text{Efficacy} = \frac{n_s}{n_t}. \tag{4}$$

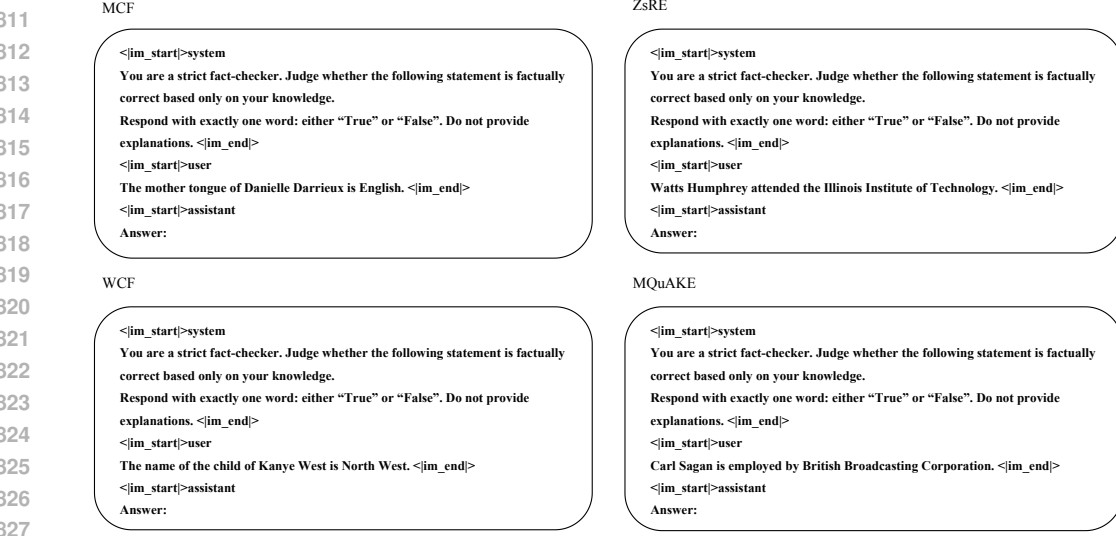

Figure 5: Examples of prompts used for fact-checking evaluation.

### A.3 THE DETAILS OF FACT-CHECKING EXPERIMENTS

For the MCF, WCF, and MQuAKE datasets, we construct statements by concatenating each query with its corresponding edit target, and then ask the model to judge whether the resulting statement is correct. The exact prompts used for each model are provided in our code repository, and illustrative examples are shown in Appendix A.5.

For the ZsRE dataset, where each query is formulated as an interrogative sentence, we concatenate the query with its answer and rewrite it into a declarative statement.

Accuracy is computed as the proportion of cases in which the model predicts "true." However, because the "old knowledge" included in these datasets does not necessarily align with the model's actual pre-edit beliefs, we excluded from the accuracy calculation those samples where the model did not answer true/false before editing, as well as those samples where the model answered "true" both before and after editing.

### A.4 THE FACT-CHECKING RESULTS WITH LLAMA MODEL

Corresponding to Table 4, the results of fact-checking with Llama are shown in Table 5.

### A.5 EXAMPLES OF PROMPT USED FOR FACT-CHECKING EVALUATION

The prompt examples used for Qwen2.5-7B-Instruct is in Figure 5. And the Llama prompts are the same except that the chat_template needs to be replaced.

### A.6 LLM USAGE

We use LLMs to refine the presentation.

### A.7 NO-EDIT BASELINE SETUP

For models that have already been edited, both the negation test and the fact-checking test can use the edit target as the ground truth. However, for the unedited model, since the edit target is unrelated to it, we need to use the old knowledge as the ground truth instead (the ZsRE does not have the old knowledge item, so we do not include the No edit baseline for it).

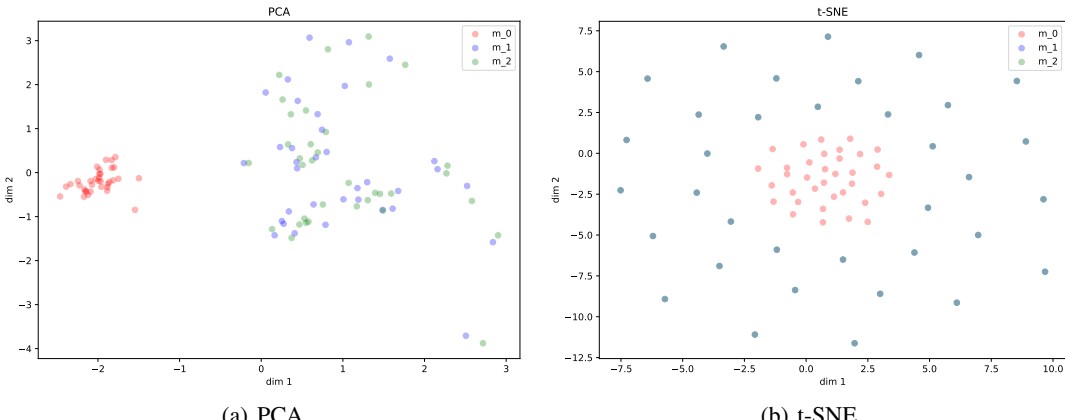

(a) PCA                                                   (b) t-SNE

Figure 6: The relationship between $m^0$ (red), $m^1$ (pruple), and $m^2$ (green). We select the eighth layer of the Llama3-8B-Instruct model. It is clear that $m^1$ is close to $m^2$, but $m^0$ are far from them. Also, $m^1$ and $m^2$ always appear as a pair. In the t-SNE plot, you can only see two types of points, this is because $m^1$ and $m^2$ are completely overlapped.

Because the old knowledge in the dataset does not necessarily align with the model's actual knowledge, we only consider samples for which the model can produce the correct output. Therefore, the efficacy metric for the unedited model is always $100\%$.

### A.8 MICROSCOPIC EXPLANATION ON WHY NEGATION DOES NOT CHANGE THE ANSWER OF EDITED MODELS

Returning to Eq. 3, we know $\Delta$ is determined by two components: $k$ and $m$. Here, $k$ is the hidden state of the decisive token before the decisive layer $W$ ($W$ is a MLP layer), and $m$ is the idealized hidden state that aligns with the edit target. We denote the output of the decisive layer at the decisive token as $l$. If we make $l = m$, then the model will finally output the edit target. And $m$ is computed through gradient descent by considering $l$ as a parameter to be optimized.

Then we will check how the $(k, m)$ pairs differ in a normal knowledge and its negation. If we get similar $(k, m)$ pairs for different knowledge, then we will get similar $\Delta$, which is an undesirable outcome.

Still using the data in Table 1 as an example:

- Old knowledge: "The mother language of Danielle Darrieux is French"

- Normal new knowledge: "The mother language of Danielle Darrieux is English"

- Negation: "The mother language of Danielle Darrieux is not English".

- Decisive token (the final token of the subject): "Darrieux".

We denote the $k$ and $m$ obtained from the normal new knowledge as $k^1$ and $m^1$, those obtained from the negation as $k^2$ and $m^2$, and those obtained from the old knowledge as $k^0$ and $m^0$ (**note that $m^0$ is directly get from the unedited model** $m^0 = l^0 = Wk^0$). In modern causal LMs, it is clear that $k^0 = k^1 = k^2$, as $k$ is determined only by the tokens before it. So, we only need to check how $m^1$ and $m^2$ differs from each other.

To do this, we analyze $m^0$, $m^1$ and $m^2$ using PCA and t-SNE in Figure 6. For clearer visualization, we selected 35 data samples (results with more data are in Figure 7) from the MCF dataset and ensured that they satisfy the following condition: for the unedited model, the model generated answers (it refers to the final answer, not the intermediate hidden state $m$) for the normal query and the negative query are completely different. We concatenate the edit target with its corresponding query to obtain $m^1$ and $m^2$.

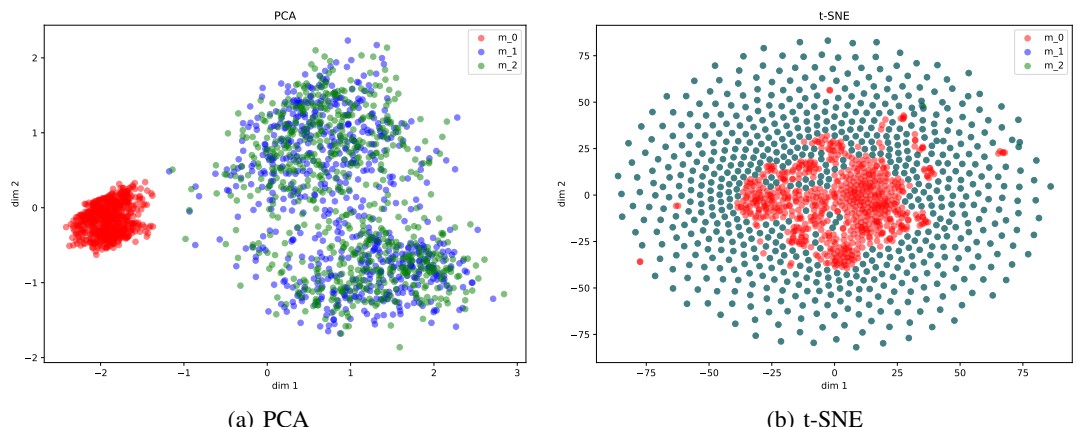

(a) PCA                                      (b) t-SNE

Figure 7: PCA and t-SNE with more data points.

Table 6: Results of original baselines with Llama3-8B-Instruct on the MCF dataset. Compared to Table 2, we remove the regularizer $\|\Delta K_p\|$ for all the baselines.

| Datasets | | MCF | | |
|---|---|---|---|---|
| Methods | | PP ↑ | PN ↓ | Discrepancy ↑ |
| | Adaedit | 0.0 | 0.0 | 0.0 |
| | EMMET | 0.0 | 0.0 | 0.0 |
| | MEMIT | 23.2 | 23.1 | 0.1 |
| Llama3-8B | NAMET | 24.8 | 24.1 | 0.7 |
| | PMET | 0.0 | 0.0 | 0.0 |
| | PRUNE | 20.8 | 20.4 | 0.4 |
| | RECT | 21.3 | 20.7 | 0.6 |
| | MEMIT-LTI | 0.1 | 0.0 | 0.1 |

The results show that $m^1$ lies close to $m^2$, whereas $m^0$ is far from them. Also, $m^1$ and $m^2$ always appear as pairs. In the t-SNE plot, only two types of points are visible because $m^1$ and $m^2$ are completely overlapping (note that t-SNE tends to compress close points into complete overlap). We note that although the outer points in the t-SNE plot appear to be widely scattered, this simply reflects that the $m^1$ vectors from different samples are far apart. For any given sample, however, its $m^1$ and $m^2$ always appear as pairs and overlap.

The results tell us that, when we steer $m^0$ towards $m^1$ with Eq 3, we are in fact also steering it towards $m^2$. Eq 3 is insensitive to the supportive tokens. Consequently, when Eq 3 is used to obtain $\Delta$, the resulting $\Delta$ carries no awareness of the distinctions contributed by those tokens.

A.9    THE RESULTS OF THE BASELINES WITHOUT OUR IMPROVED VERSION

In the AlphaEdit paper (Fang et al., 2025), it is shown that adding the regularization term $\|\Delta K_p\|$ (where $K_p$ denotes previously edited knowledge) to Eq. 3 is crucial for lifelong editing. Following this insight, we incorporate this regularizer into other baselines to improve their performance on the standard benchmark.

To ensure that our findings are not merely a consequence of adding $\|\Delta K_p\|$, we also report results for versions of the methods without this term. Since this is only a verification step, fully re-running all experiments under this setting would be computationally expensive. Instead, we conduct a focused evaluation on the most widely used dataset, MCF, using the Llama3-8B-Instruct model.

The results are presented in Table 6. We observe that the discrepancy between $PP$ and $PN$ does not improve after removing $\|\Delta K_p\|$. Instead, the $PP$ (efficacy) metric drops substantially (This drop is expected and aligns with the phenomenon shown in Table 9 of Yang et al. (2025)).

