# OpenReview forum: "Model Editing is Over: Revealing Its Illusory Success and Fragile Foundation"
_ICLR.cc/2026/Conference — Submitted to ICLR 2026_

### Official Review · Reviewer_9sBF · 2025-10-17

**Soundness:** 1
**Presentation:** 2
**Contribution:** 1
**Rating:** 2
**Confidence:** 4

**Summary:**

The paper argues that much of the reported “success” in LLM model editing stems from shortcut exploitation rather than genuine semantic integration.
It targets the standard locate-then-edit paradigm, identifies a decisive token/layer, and minimally perturbs parameters toward a target hidden state and claims this inherently incentivizes adversarial-style shortcuts over semantics. Two simple evaluations are proposed: (1) negation stress tests that combine positive/negative edit sentences with positive/negative test prompts, and (2) a fact-checking variant where the gold label is True/False rather than the edit string itself.
Across Qwen2.5-7B-Instruct and Llama-3-8B-Instruct, many editing methods (e.g., MEMIT, RECT, AlphaEdit, etc.) show very high PP efficacy but similarly high PN/NP scores, suggesting insensitivity to negation; fact-checking accuracies are much lower than PP “efficacy.” The paper concludes that current model editing rests on a fragile foundation and that evaluation should include semantically complementary negatives.

**Strengths:**

- Clear problem framing. The paper is well-written and the motivation—testing whether edits capture meaning rather than form—is intuitively strong.

- Negation and true/false checks are easy to reproduce and highlight an important gap in how we assess model editing.

**Weaknesses:**

- Overlap with existing robustness studies (limited novelty): The central claim (“model editing success is illusory under semantically perturbed queries”) has already been demonstrated in several closely related and more rigorous works, e.g., EMNLP2024 On the Robustness of Editing Large Language Models (https://aclanthology.org/2024.emnlp-main.906.pdf), prompt engineering for attacking the edits. Moreover, there is already mechanistic work going further to study why already: Revealing the Deceptiveness of Knowledge Editing: A Mechanistic Analysis of Superficial Editing (https://arxiv.org/pdf/2505.12636)

- This paper provides a useful replication and an accessible benchmark for evaluating the robustness of locate-then-edit methods, but it does not break new conceptual ground.
The related work on RAG is misleading, RAG is an inference pipeline, not a model-editing paradigm or model updating method, and more valuable discussion would instead focus on emerging non-locate-then-edit editors (hypernetwork, adapter, or inference-time). If reframed as a benchmark extension clarifying the limits of weight-space editing rather than declaring the paradigm dead, the work could become a constructive contribution to the field.

- While the experiments reveal brittleness, the rhetoric (“Model editing is over”) is scientifically excessive. A more balanced interpretation is that current locate-then-edit methods lack semantic robustness, while alternative paradigms (hypernetworks, adapters, inference-time edits) may still hold promise.

**Questions:**

none

---

> ### Author Response · Authors · 2025-11-21
> **Author rebuttal 1/2**
>
> We are deeply grateful for your thoughtful comments and constructive suggestions.
>
> **Q1. Overlap with existing robustness studies. The central claim (“model editing success is illusory under semantically perturbed queries”) has already been demonstrated in several closely related and more rigorous works ([1,2]).**
>
> **A1**. Thank you for this valuable question. We believe this concern stems from a misunderstanding. Although our work and the studies in [1,2] all examine failure cases of model editing, the findings and motivations differ substantially.
>
> The focus of [1] is the following phenomenon: when the **core semantics of a query remain the same**, but its surface form is rephrased or additional context is added, the edited model incorrectly produces different outputs. The expected behavior is that the answer should remain the same as long as the underlying semantics are unchanged. This is the notion of “robustness” explored in [1].
>
> Similarly, [2] examines cases where** the expected output should not change because the core semantics remain unchanged**. Their findings show that carefully constructed preceding contexts can make the edited model revert to the original (pre-edit) knowledge, indicating that the model has not truly “unlearned” the original fact.
>
> A commonality for the settings of [1] and [2] is that **the core semantics remain unchanged, and the gold answer is still the original edit target**.
>
>
> In contrast, our work, particularly the negation test, investigates the opposite direction. We change the semantics of the query while keeping the surface form identical, yet the edited model continues to output the same answer. In other words, while [1] and [2] document **cases where the output changes when it should stay the same**, we study **cases where the output stays the same when it should change**. These are failures in opposite directions, both in setup and in behavior.
>
> Additionally, our overarching goal is different. We argue that the high apparent efficacy of current model-editing methods stems largely from shortcut-like associations with key tokens. To reveal this limitation, we design evaluation settings in which **the gold answer is no longer the literal edit-target string**. This is the unifying principle behind both of our proposed evaluation methods.
>
>
> We apologize for omitting [1] from the related work section and have now added it.
>
>
>
> **Q2. The related work on RAG is misleading, RAG is an inference pipeline, not a model-editing paradigm or model updating method, and more valuable discussion would instead focus on emerging non-locate-then-edit editors (hypernetwork, adapter, or inference-time).**
>
> **A2**. Thank you for your suggestion. We have now revised the related work section accordingly.
>
> **Q3. If reframed as a benchmark extension clarifying the limits of weight-space editing rather than declaring the paradigm dead, the work could become a constructive contribution to the field.**
>
> **A3**. Thank you for the thoughtful suggestion. We agree that clarifying the scope strengthens the contribution. In response, we have revised both the title and the introduction to explicitly make our claims focus on locate-then-edit methods rather than the entire model-editing paradigm. We also now frame our discussion in a question-driven manner, raising concerns and highlighting limitations rather than making definitive claims about the paradigm being dead.

---

> > ### Author Response · Authors · 2025-11-21
> > **Author rebuttal 2/2**
> >
> > **Q4. A more balanced interpretation is that current locate-then-edit methods lack semantic robustness, while alternative paradigms (hypernetworks, adapters, inference-time edits) may still hold promise.**
> >
> > **A4.** Thank you for this helpful suggestion. Following your advice, we have revised both the title and the introduction to make our claims focus on the class of locate-then-edit methods. It is indeed possible that alternative paradigms may avoid the specific issues we identify. We appreciate this perspective and have narrowed our claims accordingly.
> >
> > However, locate-then-edit is currently one of the mainstream of model editing, so even with the refined scope, the implications of our findings are still substantial.
> >
> > Regarding other paradigms: our understanding is that except for inference-time editing, most alternative approaches often struggle to match the performance of locate-then-edit, and can be highly sensitive to hyperparameters and model choices. For example, two representative auxiliary-module approaches, WISE [3] and GRACE [4], completely collapse when moving from Llama-2 to Llama-3 (see issues at https://github.com/zjunlp/EasyEdit/issues/543
> > ).
> > We also attempted to include a recent hypernetwork-based method, RLEdit [5], but observed performance notably inferior to locate-then-edit.
> >
> > If you have recommendations for specific strong editing methods outside the locate-then-edit family, we would be very grateful to include them as additional baselines.
> >
> > Our re-implemented RLEdit and WISE with Llama3-8b-Instruct on the MCF dataset is as follows:
> > |         | PP   | PN   | PP-PN |
> > |---------|------|------|-------|
> > |  RLEdit | 17.8 | 15.2 | 2.6   |
> > | WISE    | 4.0  | 2.1  | 1.9   |
> >
> > RLEdit and WISE achieve low efficacy (PP), and the gap between PP and PN is also small. This may be due to some subtle configuration details not being perfectly tuned, although we have already put substantial effort into optimization. Therefore, we consider comprehensive inclusion of other method families as future work.
> >
> > We believe that inference-time editing methods are unlikely to encounter the issues we describe. They do not aggressively steer the model’s internal representations; rather, they provide additional helpful contexts.
> > Thus, we view them as a promising future direction. Nonetheless, they still inherit limitations similar to RAG, such as dependency on retrieval quality, extra latency, and increased system complexity.
> >
> >
> >
> >
> >
> > [1] On the Robustness of Editing Large Language Models. EMNLP 2024.
> > [2] Revealing the Deceptiveness of Knowledge Editing: A Mechanistic Analysis of Superficial   Editing. ACL 2025.
> > [3] WISE: rethinking the knowledge memory for lifelong model editing of large language models. NeurIPS 2024
> > [4] Aging with grace: Lifelong model editing with discrete key-value adaptors.  NeurIPS 2023.
> > [5] Reinforced lifelong editing for language models  ICML 2025.
> >
> > Thank you very much  for your careful reading and insightful feedback.

---

### Official Review · Reviewer_81Hu · 2025-10-28

**Soundness:** 1
**Presentation:** 2
**Contribution:** 1
**Rating:** 2
**Confidence:** 3

**Summary:**

This paper gives a examination of model editing, arguing that the reported success of editing methods is largely illusory. The authors claim that existing editing techniques rely on adversarial shortcuts—non-semantic correlations that enable models to output the desired edited fact without understanding or integrating it. The paper introduces two new evaluation settings: a negation test and a fact-checking test. Across multiple datasets and 2 models (Llama3-8B, Qwen2.5-7B), evaluated editing methods perform poorly under these new tests.

**Strengths:**

- The paper provides a reality check on model editing research, questioning whether its empirical progress reflects genuine knowledge integration. The connection drawn between model editing and adversarial shortcut exploitation might be useful.
- By introducing negation and fact-checking evaluations, the authors expose hidden weaknesses in editing benchmarks. These tests are conceptually simple but useful in demonstrating fragility.
- The paper evaluates major editing methods across multiple LLM architectures and datasets, offering robust empirical evidence for its caims.

**Weaknesses:**

- The central claim that “model editing is over” are exaggerated and overstated. The evidence indeed shows weaknesses in current benchmarks and methods, but limited evaluation on 2 small models does not warrant declaring the entire field invalid.
- Second, the study’s findings may not be entirely attributable to editing itself. For fair comparison, the authors should also have included baseline results for all four proposed evaluation types before editing, since some observed failures could stem from the inherent way LLMs recite or retrieve knowledge rather than the editing mechanisms.
- Third, the paper does not adequately account for the fact that LLMs are known to be highly sensitive to question format and phrasing. Including additional evaluation types, such as short-answer QA or multiple-choice questions, would provide a fairer and more comprehensive assessment of whether the observed brittleness truly arises from editing.
- the paper’s anonymous GitHub link does not correctly display or load the code,
- The authors’ claim that “supportive tokens like ‘is’ / ‘is not’ play little role at edit time” may not hold universally. This phenomenon could result from the limited reasoning and linguistic understanding capacity of smaller models such as Llama-8B, rather than a general flaw of the editing paradigm. It remains doubtful that larger, more capable frontier models would exhibit the same deficiencies.

**Questions:**

- Could you provide results for your four evaluation types (PP, PN, NN, NP, and fact-checking) before any editing is applied? This would help determine whether the observed failures stem from the editing process or from preexisting LLM limitations in handling negation and fact verification.
- Since LLMs are known to be sensitive to prompt format, did you test whether results vary when using alternative formulations, such as multiple-choice or paraphrased prompts? Additionally, do you expect the same fragility in larger models (e.g., llama-13b or llama-70b)?
- The paper attributes the observed insensitivity to “supportive tokens” (like “is” vs. “is not”) to the editing mechanism itself. Could this instead reflect the model’s limited contextual comprehension rather than the edit? A more controlled analysis isolating token-level effects would strengthen the claim.
- How do you separate the effects of editing-induced shortcuts from general weaknesses in the model’s semantic reasoning? Some of failures cases (especially in fact-checking) might reflect general LLM shortcomings rather than a specific flaw in the editing procedure.

---

> ### Author Response · Authors · 2025-11-21
> **Author rebuttal**
>
> We are grateful to your  constructive suggestions and careful evaluation of our work.
>
> **Q1 (Weakness1) The central claim that “model editing is over” are exaggerated**
>
> **A1**. Thank you for your suggestion. We agree with your assessment, and we have revised the title to make it more appropriate and balanced.
>
> **Q2 (Weakness2 & Question1) Second, the study’s findings may not be entirely attributable to editing itself. For fair comparison, the authors should also have included baseline results for all four proposed evaluation types before editing, since some observed failures could stem from the inherent way LLMs recite or retrieve knowledge rather than the editing mechanisms.**
>
> **A2**. We are grateful for your valuable suggestion. This is indeed an important concern. To address it, we have added the vanilla (unedited) model as an additional baseline in all experiments (Tables 2–5). We observe that the unedited model performs well on our specially designed tasks. Therefore, the failures of the edited models do not primarily stem from the base model’s inability to understand these queries.
>
>  **Q3 (Weakness3,5 & Question2,3,4) Third, the paper does not adequately account for the fact that LLMs are known to be highly sensitive to question format and phrasing. Including additional evaluation types, such as short-answer QA or multiple-choice questions, would provide a fairer and more comprehensive assessment of whether the observed brittleness truly arises from editing.**
>
> **A3**  Thank you very much for this insightful comment. Although we are currently unable to run experiments on Llama-13B or Llama-70B, we believe the following clarification may help address your concern.
>
> As you know, model editing relies critically on identifying the decisive layers of a model. This step is essential for achieving strong editing performance. However, locating these layers via causal tracing is computationally expensive. In our experiments, we rely on the public EasyEdit repository, which provides layer-location annotations for several mainstream models. Unfortunately, no such results exist for models of 13B parameters or larger, and conducting this procedure ourselves would require substantial data and computational resources.
>
> Nonetheless, we can address your concern from another perspective. Empirically, we include the results of unedited models for both the negation tests (Tables 2–3) and the fact-checking tests (Tables 4–5). The unedited models exhibit much lower hallucination rates than the edited models. For example, Llama3-8B shows an average hallucination rate of only 9.2% (Table 2). Likewise, the vanilla models continue to perform well in the fact-checking task (Table 4). These results demonstrate that the base model is capable of understanding our queries. Thus, the brittleness is introduced mainly by editing, not by intrinsic limitations of the original model.
>
> And from a philosophical perfpective, our findings are beyond the issue of "LLMs are highly sensitive to question format and phrasing". Generally, the problem "LLMs are highly sensitive to question format and phrasing" is understood as the model producing **different outputs** when the question’s **format is slightly perturbed** while **its semantics remain unchanged**.
> In contrast, our negation test goes in the opposite direction: **we change the semantics** of the query while keeping the format unchanged, yet the edited model still **outputs the same answer**.
> Although both are forms of model failure, they arise from fundamentally different causes and exhibit opposite behaviors.
>
> Therefore, both empirically and conceptually, our findings go beyond the problem of the vanilla model’s ability to understand the test queries.
>
> **Q4 (Weakness 4) the paper’s anonymous GitHub link does not correctly display or load the code.**
>
> **A4.** We apologize for this oversight. We have verified the link (https://anonymous.4open.science/r/me-1DF4), and it now displays correctly. The earlier issue was likely due to temporary network or permission problems.
>
> **Q5 (Question4) How do you separate the effects of editing-induced shortcuts from general weaknesses in the model’s semantic reasoning? Some of failures cases might reflect general LLM shortcomings rather than a specific flaw in the editing procedure.**
>
> **A5.** To separate the effects of editing-induced shortcuts from the model’s inherent reasoning limitations, we have included results from the vanilla (unedited) model in all relevant experiments. The unedited model does exhibit a non-zero failure rate; however, its error rate is consistently much lower than that of the edited models across both the negation tests and the fact-checking tasks. This indicates that although a small portion of the failures may stem from the underlying model’s general weaknesses, the majority clearly arise from the editing procedures themselves rather than from an inherent inability of the base model to perform these tasks.

---

> > ### Comment · Reviewer_81Hu · 2025-11-27
> >
> > The authors have partially addressed some of the issues, and I have raised my score accordingly. However, the response would be more convincing with supporting evidence for weaknesses 3 and 5.

---

### Official Review · Reviewer_eq5o · 2025-10-29

**Soundness:** 2
**Presentation:** 3
**Contribution:** 2
**Rating:** 4
**Confidence:** 4

**Summary:**

This is an interesting study that raises fundamental questions about the mainstream evaluation methods and core mechanisms in the field of Large Language Model (LLM) knowledge editing. The authors use simple yet ingenious experiments (negation queries and fact-checking) to compellingly demonstrate that existing SOTA methods rely primarily on "adversarial shortcuts" rather than genuine semantic knowledge integration. The experiments show surprising results for the current methods.

**Strengths:**

1. The paper addresses a crucial, long-overlooked defect in the model editing literature—the lack of "negative case" evaluation. It boldly challenges the reported success of mainstream methods, pointing the way toward more robust directions for future research in the field.

2. The proposed "Simple Negation Test" (PN/NP) and "Fact-Checking Style Evaluation" are interesting and useful. These methods are simple in design but effective at exposing severe deficiencies in the semantic completeness and robustness of current methods.

3. The paper extensively validates its claims across two mainstream LLMs (Llama-3-8B-Instruct, Qwen2.5-7B-Instruct) and nine SOTA editing methods, ensuring the universality of its conclusions.

**Weaknesses:**

1. In the fact-checking experiments, the model switches from generating facts (the original knowledge editing task) to judging truthfulness (the new task). Does this task switching itself introduce confounding factors? Although the authors state that "the two evaluation tasks are roughly comparable in difficulty," it might be worth further discussion or including a control experiment to rule out the influence of task conversion on the results, thereby ensuring the performance drop is solely attributable to the failure of semantic integration.

2. Although the proposed insight is interesting, the paper does not attempt to solve this problem or discuss how to solve this problem. And the title is kind of histrionic or slightly aggressive. Given that this paper aims to advance the field, it is suggested that the conclusion section be made more constructive

3. Although the paper provides a strong analogy, a deeper mechanistic analysis is needed regarding why the "locate-then-edit" optimization objective (Eq. 3) necessarily leads to this shortcut behavior. For instance, why does intervention on the decisive token's hidden state actively neglect supportive tokens in the context (e.g., "is/is not")? Providing a microscopic explanation based on gradients or attention mechanisms would significantly strengthen the paper's foundation.

**Questions:**

Q1: What can the Discrepancy in Tables 2 and 3 indicate? It seems the metric could not reveal any insights.

Q2: The author claims they implement them with our improved version. What about the performance of the original methods in the evaluation? It is not very convincing.

---

> ### Author Response · Authors · 2025-11-21
> **Author rebuttal 1/2**
>
> We sincerely thank you for the valuable feedback and thoughtful comments.
>
> **Q1. The difficulty of fact-checking experiment.**
>
> **A1.**
> Thank you for your suggestion, and we agree that clarification is needed. Intuitively, for semantically aware LLMs, the fact-checking task should be easier because the possible answer is already present in the context, and the model only needs to determine whether the statement is true or false. In contrast, the generative task requires the model to produce the entire correct answer (a sequence of multiple tokens) on its own.
>
> We also empirically validate this point. We add the results of the unedited model to all our experiments (Tables 2–5), and the detailed setting for this baseline is in Appendix A.7. We observe that the unedited model performs well on the fact-checking task, with all accuracies exceeding 90%.
>
>
>
> **Q2. The title is kind of histrionic or slightly aggressive**.
>
> **A2**. Thank you very much. We agree with your assessment, and we have revised the title to make it more appropriate and balanced. We also narrowed the scope so that the title better reflects the objective nature of our evaluation.
>
> **Q3. The paper does not attempt to solve this problem or discuss how to solve this problem**.
>
> **A3.** Thank you for your suggestion. We agree that proposing solutions would be valuable. However, as clarified in the paper, the issue we identify is tightly coupled with the fundamental objective of locate-then-edit methods (steering the model toward a target output using minimal interventions). This makes the problem inherently challenging, and we currently do not have an effective solution.
>
> Our main contribution belongs to negative findings and designing evaluation methods that better reveal this issue, rather than proposing techniques to address it.
>
> We think that more steady and balanced approaches, such as RAG style inference-time knowledge editing, may mitigate this problem because they do not aggressively alter the model’s internal representations; instead, they provide additional contextual information.
> Fine-tuning with data augmentation may also alleviate the issue, as complementary knowledge can be absorbed from multiple training samples. However, these methods also have limitations, including reliance on retrieval quality, additional latency and system complexity (for inference-time editing), and the need for high-quality data (for fine-tuning). In future work, beyond the locate-then-edit paradigm, we plan to explore improvements to these alternative approaches as well.
>
> We appreciate your suggestion and have added this discussion to the conclusion section.
>
> **Q4. Why does intervention on the decisive token's hidden state actively neglect supportive tokens in the context. Providing a microscopic explanation based on gradients or attention mechanisms would significantly strengthen the paper's foundation.**
>
> **A4**. Thank you for your suggestion. We have added a more detailed analysis of why this problem occurs in Appendix A.8 (Figure 6). Specifically, we analyze the optimization objective in Eq. 3 and examine the differences in the $(k,m)$ pairs between normal knowledge and its negated counterpart.
> First, in causal LMs, $k$ is theoretically unaffected by the tokens after it. Then, by applying PCA and t-SNE to examine $m$ under different conditions, we find that the directions of
> $m$ in both cases are extremely similar.
>
> Consequently, when Eq. 3 is used to obtain $\Delta$, it lacks awareness of the supportive tokens.
>
> We can also find clues from the earliest locate-then-edit paper, ROME [1]. In Fig. 1 (causal tracing), decisive layers show that only the decisive tokens are strongly correlated with the target output, while other tokens in those layers exhibit much weaker correlations.
>
>  **Q5. What can the Discrepancy in Tables 2 and 3 indicate?**
>
>  **A5**. This quantity can be viewed as a form of rectified accuracy. We include it to avoid the following ambiguity: suppose method A achieves both high Efficacy and high Hallucination, while method B obtains low scores on both metrics, how should one compare them?
>
> As you know, high Hallucination is often a byproduct of high Efficacy. Reporting only one of the metrics may allow methods to exploit loopholes. Thus, the two metrics are tightly entangled and should be considered together. To enable a unified metric that captures both aspects, we designed the Rectified Efficacy. At the same time, directly examining the discrepancy provides an intuitive way to assess the extent to which supportive tokens contribute.

---

> ### Author Response · Authors · 2025-11-21
> **Author rebuttal 2/2**
>
> **Q6  The author claims they implement them with our improved version. What about the performance of the original methods in the evaluation? It is not very convincing.**
>
>  **A6**. Thank you for your suggestion. We have added the original versions of these methods to Table 6 in Appendix A.9. We observe that the discrepancy between $PP$ and $PN$ does not improve even when the $||\Delta K_p||$ term is removed. Instead, the efficacy score drops significantly.
>
> In the AlphaEdit [2] paper , it is shown that adding
> $||\Delta K_p||$ (where $K_p$ denotes previously edited knowledge) into Eq. 3 as a regularizer is important for lifelong editing. Therefore, we applied this technique to other baselines to improve their performance on the standard benchmark.
>
>  [1] Locating and Editing Factual Associations in GPT.
>  [2] Alphaedit: Null-space constrained knowledge editing for language models.
>
> Thank you for your valuable time and insights.

---

### Official Review · Reviewer_g7i2 · 2025-11-02

**Soundness:** 2
**Presentation:** 3
**Contribution:** 2
**Rating:** 4
**Confidence:** 2

**Summary:**

This paper presents a profound and critical challenge to the foundations of the rapidly growing field of model editing for Large Language Models (LLMs). The authors argue that the apparent high success rates of current model editing techniques, as measured by established benchmarks, are largely illusory and built upon a fragile foundation. Their central thesis is that the core objective of model editing—to steer the model's output to a target with minimal parameter changes—inherently encourages the model to learn "adversarial shortcuts." This means the model forms a superficial association between a trigger pattern (e.g., the subject token) and the target answer, bypassing a genuine understanding and integration of the knowledge's full semantics.

To substantiate this claim, the authors introduce a novel evaluation framework that systematically incorporates ​negative cases. This includes:
1. ​Simple Negation Queries: Testing edited models with negated versions of the original query (e.g., "XX is not" instead of "XX is"). Strikingly, the models still confidently output the edit target, demonstrating a failure to comprehend the logical negation.
​2. Fact-Checking Evaluation: Requiring the model to judge the truthfulness of a statement containing the edited fact, rather than directly generating it. This reveals a significant performance drop compared to standard generation-based evaluation.
Through extensive experiments on two base LLMs (Llama3-8B and Qwen2.5-7B) involving nine state-of-the-art editing methods across four standard datasets, the paper provides compelling evidence. The results consistently show that all methods collapse under negation queries and perform poorly on fact-checking, strongly supporting the authors' contention that current editing paradigms rely on shortcuts rather than robust semantic integration. The paper concludes by calling for a fundamental re-examination of the field's evaluation practices and underlying assumptions.

**Strengths:**

1 ​Paradigm-Challenging Perspective: The paper successfully reframes model editing as a potential instance of adversarial shortcut learning, providing a new lens through which to evaluate editing techniques.
​2 Methodological Contribution: The proposed negative-case evaluation framework addresses a critical gap in current benchmarking practices and sets a new standard for robustness assessment.
3 ​Rigorous Experimental Design: The comprehensive evaluation across methods, models, and datasets ensures the findings are generalizable and not method-specific.

**Weaknesses:**

​1. Mechanistic Explanation: The paper demonstrates the existence of shortcuts but lacks a detailed analysis of their internal mechanisms. For example, do edits primarily alter attention patterns in specific layers or disrupt logical operations (e.g., negation handling) in feedforward networks? Incorporating neuron-level analyses (e.g., causal tracing post-edit) could clarify how shortcuts manifest.
2. ​Evaluation Confounders: The negation-based tests assume LLMs can inherently handle negation, but baseline performance on negation tasks is not benchmarked. If vanilla models struggle with negation, the editing-specific failure may be overstated. A control experiment testing negation understanding in unedited models would strengthen causality.
​3. Paradigm Boundaries: The critique focuses on "locate-then-edit" methods but does not dissect how alternative approaches (e.g., hypernetworks or external modules) might avoid these pitfalls. Clarifying whether the issue is paradigm-specific or universal would refine the paper’s scope.
4. ​Practical Implications: The experiments use simplified settings; assessing whether shortcuts harm real-world tasks (e.g., multi-hop reasoning post-edit) would amplify the work’s applicability.

**Questions:**

1. Could you elaborate on the analogy between model editing and adversarial attacks? Specifically, how do shortcuts in parameter space(editing) differ from those in input space(attacks), and does this suggest unique mitigation strategies?
2. The results show consistent output of the edit target across all query types. Does this imply that edits weaken the model’s semantic understanding of predicates (e.g., "is" vs. "is not")? Is there evidence of degraded logical reasoning post-edit?
3. How might future editing paradigms balance precision and semantic completeness? For instance, could incorporating negative examples during editing or using logic-based constraints help?
4. Does the failure under negation queries generalize to more complex logical forms (e.g., quantifiers like "never" or "always")?

---

> ### Author Response · Authors · 2025-11-21
> **Author rebuttal 1/2**
>
> We sincerely thank you for your constructive input and encouragement.
>
> **Q1 (Weakness1 & Question1) Lack of Mechanistic Explanation**.
>
> **A1**. Thank you very much for your suggestion. We have now added a more detailed analysis of why this problem occurs in Appendix A.8 (Figure 6). Specifically, we analyze the optimization objective in Eq. 3 and examine differences in the $(k,m)$ pairs between the normal knowledge and its corresponding negation.
> First, in causal LMs, $k$ is theoretically unaffected by negation. Then, by applying PCA and t-SNE to examine $m$ under different conditions, we find that the directions of $m$ in these two cases are very similar.
>
> Therefore, when we use Eq. 3 to compute $\Delta$, it fails to incorporate awareness of the supportive tokens.
>
> We can also see some clues from the earliest locate-then-edit paper, ROME [1]. In Fig. 1 (causal tracing) of [1], the decisive layers show that only decisive tokens are highly correlated with the target output, while other tokens in those layers have much weaker correlations.
>
>
>  **Q2 (Weakness2 & Question2) ​Evaluation Confounders: The negation-based tests assume LLMs can inherently handle negation, but baseline performance on negation tasks is not benchmarked. If vanilla models struggle with negation, the editing-specific failure may be overstated.**
>
>  **A2**. We are grateful for your valuable suggestion. This is indeed an important concern. To address it, we have added the results of the vanilla (unedited) model as an additional baseline in all experiments (Tables 2–5). We observe that the unedited model performs well on our specially designed tasks. For example, unedited Llama3-8B exhibits an average hallucination rate of only 9.2% (Table 2). Likewise, in the fact-checking task, the vanilla models also show strong performance (Table 4). These results demonstrate that the model itself is capable of understanding our queries.
>
> Thus, the brittleness is introduced mainly by editing, rather than by intrinsic limitations of the original model.
>
>  **Q3 (Weakness3) Paradigm Boundaries: The critique focuses on "locate-then-edit" methods but does not dissect how alternative approaches (e.g., hypernetworks or external modules) might avoid these pitfalls.**
>
>  **A3**. Regarding other paradigms: our understanding is that except for inference-time editing, most alternative approaches often struggle to match the performance of locate-then-edit, and can be highly sensitive to hyperparameters and model choices. For example, two representative auxiliary-module approaches, WISE [1] and GRACE [2], completely collapse when moving from Llama-2 to Llama-3 (see issues at https://github.com/zjunlp/EasyEdit/issues/543
> ).
> We also attempted to include a recent hypernetwork-based method, RLEdit [3], but observed performance notably inferior to locate-then-edit.
>
> If you have recommendations for specific strong editing methods outside the locate-then-edit family, we would be very grateful to include them as additional baselines.
>
> Our re-implemented RLEdit and WISE with Llama3-8b-Instruct on the MCF dataset is as follows:
> |         | PP   | PN   | PP-PN |
> |---------|------|------|-------|
> |  RLEdit | 17.8 | 15.2 | 2.6   |
> | WISE    | 4.0  | 2.1  | 1.9   |
>
> RLEdit and WISE achieve low efficacy (PP), and the gap between PP and PN is also small. This may be due to some subtle configuration details not being perfectly tuned, although we have already put substantial effort into optimization. Therefore, we consider comprehensive inclusion of other method families as future work.
>
> We believe that inference-time editing methods are unlikely to encounter the issues we describe. They do not aggressively steer the model’s internal representations; rather, they provide additional helpful contexts.
> Thus, we view them as a promising future direction. Nonetheless, they still inherit limitations similar to RAG, such as dependency on retrieval quality, extra latency, and increased system complexity.
>
> [1] WISE: rethinking the knowledge memory for lifelong model editing of large language models. NeurIPS 2024
> [2] Aging with grace: Lifelong model editing with discrete key-value adaptors.  NeurIPS 2023.
> [3] Reinforced lifelong editing for language models  ICML 2025.

---

> ### Author Response · Authors · 2025-11-21
> **Author rebuttal 2/2**
>
> **Q4 (Weakness4 & Question 4) ​Practical Implications: The experiments use simplified settings; assessing whether shortcuts harm real-world tasks (e.g., multi-hop reasoning post-edit) would amplify the work’s applicability.**
>
> **A4**. Thank you for your suggestion, but we think this is a misunderstanding. Our goal is not to develop new methods that improve model performance, but rather to expose weaknesses in existing editing approaches. In this context, using simplified settings is actually an advantage, not a limitation.
>
> Our results show that current methods already fail under very simple evaluation setups (e.g., fact-checking). If a method cannot remain reliable in minimal and controlled conditions, its performance is unlikely to improve, and will almost certainly degrade, on more complex, real-world tasks such as multi-hop reasoning.
>
> Therefore, our simplified settings should be viewed as a stress test that reveals the inherent brittleness of current techniques, rather than an oversimplification that limits applicability.
>
> **Q5 (Question3) How might future editing paradigms balance precision and semantic completeness? For instance, could incorporating negative examples during editing or using logic-based constraints help?**
>
> **A5**. Thank you for the valuable question. We think that more steady and balanced approaches, such as RAG style inference-time knowledge editing, can avoid this problem. Because they do not aggressively steer the output; instead, they simply provide helpful context.
> And fine-tuning the model with data augmentation, might also avoid this problem because complementary knowledge can be absorbed from multiple data samples. However, they also face their own limitations, such as dependence on retrieval quality, additional latency, system complexity (for inference-time editing), and lack of high-quality data (for fine-tuning). In the future, besides the locate-then-edit approach, we may consider giving more attention to improving these methods as well.
>
> We are grateful for your suggestion and have added this discussion into the conclusion section.

---

### Author Response · Authors · 2025-11-21
**Summary of our rebuttal and the corresponding revisions**

We sincerely appreciate the constructive feedback provided by all reviewers.

Below is a summary of our rebuttal and the corresponding revisions:

1. Revising the title and introduction to make more focused scope.
Following the reviewers’ suggestions, we have revised the paper's title in the submitted PDF (OpenReview does not currently offer functionality for updating the title at rebuttal time). We have also updated the abstract and introduction to ensure that our claims focus on the locate-then-edit class of model-editing methods.

2. Adding the unedited model as an additional baseline to rule out prompt-sensitivity effects.
Several reviewers raised an important concern: the observed failures may stem from limitations of the unedited (vanilla) model rather than from the editing process itself. To address this, we have added the performance of the unedited model as an additional baseline across all relevant experiments (Tables 2–5). We find that the unedited model performs well on our designed tasks, confirming that the failures of the edited models do not originate from deficiencies in the underlying model’s ability to understand the queries.

3. Providing a more detailed mechanistic explanation.
We have expanded our analysis and added a more thorough mechanistic explanation of why the issue arises. This detailed discussion is Appendix A.8.

---

### Meta-Review · Area_Chair_zpiy · 2026-01-03

**Summary:**

Reviewers raised several key concerns that highlight limitations in the paper's scope, evaluation rigor, and novelty, leading to a recommendation of marginal rejection (borderline, with potential for revision): (1) insufficient mechanistic analysis of why shortcuts emerge post-editing, lacking neuron-level insights or causal tracing (Reviewer g7i2); (2) potential confounders in evaluations, such as unbenchmarked negation handling in vanilla models, task-switching effects in fact-checking, and sensitivity to prompt phrasing without controls like multiple-choice formats (Reviewers g7i2, eq5o, 81Hu); (3) overstated claims and narrow focus on locate-then-edit paradigms without dissecting alternatives (e.g., hypernetworks, RAG) or broader real-world applicability (Reviewers g7i2, 81Hu, 9sBF); and (4) limited novelty due to overlap with prior robustness studies on model editing, which are more comprehensive (Reviewer 9sBF). While the rebuttal addresses some issues (e.g., adding baselines), lingering gaps in generalizability and evidence for non-editing paradigms weaken the overall contribution.

**Reviewer Concerns:**

The rebuttal addressed key concerns including adding unedited model baselines to rule out inherent LLM limitations, providing a mechanistic analysis in Appendix A.8, revising the title and scope to focus on locate-then-edit methods, and discussing alternative paradigms like inference-time editing. Outstanding concerns include potential overlap with prior work (distinguished but not fully resolved), lack of experiments on larger models or additional formats like multiple-choice questions, and limited evidence for real-world implications or solutions to the identified issues.

**Reviewer Scores:**

Overall, reviewers are likely to maintain or slightly increase their scores, as the rebuttal partially addresses several concerns but does not fully resolve all of them, including the need for results on additional prompt variations and more comprehensive clarification of overlaps with prior work or scope limitations.

---

### Decision · Program_Chairs · 2026-01-26

Reject